# Effect of PLA/PBAT Antibacterial Film on Storage Quality of Passion Fruit during the Shelf-Life

**DOI:** 10.3390/molecules24183378

**Published:** 2019-09-17

**Authors:** Rong Zhang, Wenting Lan, Jie Ding, Saeed Ahmed, Wen Qin, Li He, Yaowen Liu

**Affiliations:** 1College of Food Science, Sichuan Agricultural University, Yaan 625014, China; zhangronglzy@163.com (R.Z.); 18227593253@163.com (W.L.); saeedahmed_mahar@yahoo.com (S.A.); qinwen@sicau.edu.cn (W.Q.); 2College of Food Science and Technology, Sichuan Tourism University, Chengdu 610100, China; dingjiedream@163.com; 3School of Materials Science and Engineering, Southwest Jiaotong University, Chengdu 610031, China; 4California NanoSystems Institute, University of California, Los Angeles, CA 90095, USA

**Keywords:** passion fruit, PLA/PBAT antibacterial films, storage quality

## Abstract

In this experiment, we studied the effect of poly(lactic acid)/poly(butylene adipate-co-terephthalate) (PLA/PBAT) blend films on the efficiency of passion fruit preservation at 20 °C. The weight loss, shrinkage index, firmness, and total sugar of passion fruit packaged with PLA/PBAT films had no significant differences compared with PE films during 21 days (*p* > 0.05). PLA/PBAT films can more effectively reduce the rising of ethanol content and delay the total acid, ascorbic acid, and sensory evaluation. Compared with unpackaged (CK) and polyethylene (PE) films, PLA/PBAT films are more conducive to preserve the overall flavor of passion fruit during storage time, in agreement with sensory evaluation, tested by E-nose, E-tongue, and GC-MS, which also proved that it can effectively maintain the edible quality of passion fruit during storage time. We believe that our study makes a significant contribution to literature because it paves the way to the generalization and application of packaging films based on composite antibacterial polymers and facilitates the commercialization of fresh passion fruit as an important health food.

## 1. Introduction

Purple passion fruit (*Passiflora adults Sims*) is a perennial evergreen climbing vine, which carries ripe fruit between August the same year and March the next year [1]. The ripened fruit, called “fruit monosodium glutamate”, tends to have bright color, unique and rich aroma, whose juice can emit complex aroma of pomegranate, pineapple, strawberry, and other fruits [2]. In recent years, passion fruit, characterized by distinct flavor and ample nutrition is a standing and popular product that is targeted to the Chinese fresh fruit market. Motivated by vigorous market demand and backed up by the government policy, the passion fruit planting area has been strengthened year by year in China. Now, purple passion fruit production is an important source of income for small-holder farmers in the poverty-stricken districts of southern China province. However, high temperature in harvesting season makes passion fruit, as a typical kind of respiratory climactic fruit, highly susceptible to water loss, which easily leads to fruit shrivelling and causes a reduction in juice quality and physiological deterioration, to produce a peculiar smell during storage, therefore resulting in a short shelf-life [3]. In order to reduce the loss of postharvest, many researchers focused on cheaper and more effectual preservative methods.

Many studies have been focused on nutritional composition as well as medicinal potential of *Passiflora foetida L*. plant extracts and passionflower fruit juice processing [4]. The preservation techniques mainly related to calcium treatment, hot-water treatment, 1-Methylcyclopropene (1-MCP) treatment, and modified atmosphere packages. A. Baraza et al. suggested that commercial application of 1-MCP in passion fruit at turning stage (60–65 days after anthesis) was optimal to ensure efficacy, since they were still marketable on day 14 [5]. Jacqueline Barbosa Dutra et al. found that the combination of hot-water treatment at 47 °C (4 or 5 min) and application of the phosphite of K or Zn could significantly reduce the disease severity in fruits; the 1-MCP treatment reduced the anthracnose severity in passion fruit mainly at 200 nL L^−1^ · 24 h^−1^ [6]. Da Silva et al. studied that the application of CaCl_2_ by infiltration favored the maintenance of fruit consistency during the first 18 days of storage time [7]. Maniwara, P. et al. used modified atmosphere packaging (MAP) with an oxygen transmission rate of 12,000 cm^3^/m^2^∙day∙atm and showed the best result in maintaining fruit quality, gas composition, and extension of storage life (up to 51 days) [8]. The high potential of the atmosphere modification technique and the growing interest in large-scale retail trade in fresh berry fruit has drawn increased attention to the development of novel food packaging techniques, particularly those avoiding environmental pollution with plastic via the use of biodegradable bio-based polymers [9,10,11].

Covered passion fruit with polyethylene (PE) film packaging is widely used in China market, because the packaging enhances the shelf-life and quality of passion fruits [12,13]. However, it is difficult to degrade and easily leads to serious plastic pollution. The market urgently needs the products with the same fresh-keeping effect but environment-friendly packaging to replace PE fresh-keeping films. Lactic acid-based polymers are known as poly(lactic acid) (PLA), which has high mechanical properties, thermal plasticity, and fabric ability, and these monomers are getting more and more attention [14]. Some previous studies were performed to improve performance properties (such as impact strength, flexibility, stiffness, gas barrier properties, and thermal stability) of PLA, using addition of modifiers, blending, compatibilization, and physical treatments to enhance the quality of stored food [15]. In order to attend to the growing consumer demand for innovative ready-to-eat fresh, cut fruit packaging, PLA-based packages were widely investigated [16]. PLA/Ag/Vitamin E membrane actively reduced the polyphenol oxidase activity, which could have application in the food industry as a potential preservative packaging for fruit and juice [17]. The flexibility of PLA film would be increased significantly by blending a small amount of poly(butylene adipate-co-terephthalate) (PBAT), and it was convenient to retain the freshness of packaged green onion [18]. Although PLA/PBAT blend film is well known for exhibiting excellent physical properties, its use to preserve passion fruit has not been extensively reported. Present reports of using the electronic tongue and nose on the effect of different packaging films on the edible quality of passion fruit during the shelf-life are limited.

Therefore, this work aimed to study the effect of PLA/PBAT blend films on the efficiency of passion fruit preservation at 20 °C; nonpackaged and PE films was used as control groups. This study also aimed to characterize the volatile compounds responsible and taste profile for the flavor of each state of passion fruit ripeness, using electronic tongue and nose.

## 2. Results and Discussion

### 2.1. Weight Loss

Passion fruits are climacteric fruits which exhibit a short postharvest life due to rapid water loss and shriveling, which contributes to loss in marketability [5]. Weight loss was mainly ascribed to respiration and moisture evaporation through the passion fruit skin and resulted in fruit shrinkage and deterioration. The water-loss rate was determined by the water pressure gradient between fruit and the surrounding atmosphere and storage temperature [19]. Figure 1a shows the weight loss of passion fruits during 21 days room-temperature storage, revealing that all samples exhibited a continuous weight loss with time, with the greatest loss observed for the unpackaged (CK) sample. The weight loss of the CK sample after twelve days reached 11.52%, which was significantly higher than that of packaged samples (*p* < 0.05), since it was primarily caused by the migration of water from the fruit to the environment [20]. PLA/PBAT and PE packaging treatments can all effectively reduce the weight loss of passion fruits compared with the unpackaged sample. For packaged samples, the presence of packaging materials semipermeable to carbon dioxide, oxygen, and moisture decreased the extents of respiration, water loss, and oxidation, which was reflected in decreased weight loss [21]. Though PLA/PBAT films had higher oxygen permeability and water vapor permeability than PE, no significant difference in the weight loss was observed between the two packaged treatments. So, PLA/PBAT can effectively delay the descending rate of fruit weight loss compared with PE.

### 2.2. Shrinkage Index

Fruit shrinkage of peel is one of the important indicators to evaluate the storage quality of fruit, which can seriously affect the appearance quality of fruit, thus affecting their commercial value. Figure 1b shows there were no significant differences in these treatments at earlier storage (from 1 to 3 days). The shrinkage index of all samples trended towards an increase in the subsequent process of storage (from 6 to 21 days). During the storage period, the CK sample was observed with the greatest shrinkage index., which was significantly higher than those of packaged samples (*p* < 0.05). The shrinkage index of the CK sample after 12 days reached 3.07, thus the commodity value was less. The shrinkage of PE and PLA/PBAT treatments increased slowly during the first 9 days compared with CK, and rapidly exceeded 3.0 until the 18^th^ day. However, no significant difference in the shrinkage index was observed between the two packaged treatments (*p* > 0.05). There were great differences between the PLA/PBAT and PE groups at the 9th day (*p* < 0.05), but they had a similar tendency in the subsequent process of storage (from 12 to 21 days), when the two groups showed no significant difference (*p* > 0.05). Thus, for reducing the shrinkage index of passion fruit in storage, PPLA/PBAT and PE packaging films will have the same effect.

### 2.3. Firmness

Firmness had excellent correlation to storage quality of fruit and vegetables, and can take into consideration the process of ripening and senescence in fruit. Figure 1c shows the firmness of passion fruit during 21 day room temperature storage, revealing that all samples exhibited a continuous drop with time, with the greatest loss observed in the CK sample. Firmness values measured at earlier storage (from 1 to 6 days) were similar for all samples (*p* > 0.05) and began to be varied after nine days (*p* < 0.05; the firmness of samples significantly decreased after six (unpackaged) and nine (PLA/PBAT and PE) days of storage. After the 9th day of storage, firmness losses of 41.61%, 10.40%, and 19.51% were found in unpackaged, PE- and PLA/PBAT-packaged passion fruit, respectively (i.e., packaging helped to retain firmness). The PLA/PBAT group had no significant differences in comparison with the PE group during the latter stage of storage (from 12 to 21 days) (*p* > 0.05). Thus, PLA/PBAT can more effectively delay the descending rate of fruit firmness compared with PE.

### 2.4. Total Sugar

The quality and consumer perception of fruit is strongly affected by total sugar. Figure 1d presents the results of total sugar analysis during 21 d storage, revealing that total sugar of all groups increased first and then decreased, then the total sugar peaks of CK and packaging groups were reached at the 12th and 15th day, respectively. During this period, the increase rate of the CK group was significantly greater than that of other treatments. Starting from day 15, the sharpest increase of total sugar was observed for PLA/PABT- and PE-packaged samples (8.25% and 8.55%, respectively), but the difference between the two groups was not evident, while the total sugar in the CK group dropped to about 6.55% at the same time. After the peak of total sugar appeared, the PLA/PBAT group had no significant differences in comparison with the PE group during the latter stage of storage (from 15 to 21 days) (*p* > 0.05). Thus, PLA/ PBAT can also effectively reduce the senescence of passion fruits and delay the appearance of total sugar peak compared with PE.

### 2.5. Total Acid

After soluble sugar, organic nonvolatile acids are the second most significant contributor to fruit flavor. Respiration, resulting in untreated acid consumption and fruit senescence, is thought to be one of the main reasons for total acid (TA) changes [22]. Figure 2a shows the effect of packaging on passion fruit total acid content, demonstrating that in all cases, TA decreased with time during storage, with the slowest decrease observed for fruit packaged in PLA/PBAT film. This result demonstrated that packaging materials can delay the utilization of organic acids and hence slow down the TA reduction, in agreement with our previous report [23]. In particular, the TA decrease (49.83 to 29.45 (1 g citric acid kg^-1^) over 21 d) observed for PLA/PBAT-packaged fruit was higher than that observed for PE-packaged samples, which was ascribed to the antibacterial effect of PLA/PBAT and its ability to slow down the deterioration of fruit during storage. Thus, packaging materials significantly (*p* < 0.05) reduced the rate of TA reduction. These were signs that PLA/PBAT can more effectively reduce the senescence of passion fruit and delay the total acid reduction than PE. This proves that PLA/PBAT can effectively maintain the edible quality of passion fruit during storage.

### 2.6. Ascorbic Acid Content

Figure 2b displays the ascorbic acid content of passion fruit in all treatment groups, which showed a downward trend. After storage, the ascorbic acid content of all samples decreased (*p* < 0.05), with the lowest value observed for the unpackaged sample (12.65 ± 0.14 (mg/100 g) at 21 d) and the highest value observed for the PLA/PBAT-packaged sample (15.00 ± 1.12 (mg/100 g) at 21 d). Ascorbic acid content measured at earlier storage (from 1 to 9 days) was similar for all packaged samples (*p* > 0.05) and began to get varied after nine days (*p* < 0.05), with that of the PE group significantly decreased compared with the PLA/PBAT group. Thus, PLA/PBAT can more effectively delay the ascorbic acid oxidizing than PE. Ascorbic acid of fruit was best preserved in the case of PLA/PBAT-packaged samples. In particular, the score decrease (25.8 ± 1.83 to 9.40 ± 1.60 over 21 d) observed for PLA/PBAT-packaged fruit was lower than that observed for PE-packaged samples, which was ascribed to high gas permeability of PLA/PBAT and its ability to slow down the ethanol production of fruit during storage.

### 2.7. Ethanlo Content

The high-density peel serves as a barrier that obstructs gas permeability into the fruit’s internal cavity. Anaerobic respiration produces acetaldehydes and ethanol as it ferments [8]. During the experiment, ethanol concentration in passion fruit markedly rose in the storage period in all groups (Figure 2c). Ethanol reached up to 0.42 g/100 g within 9 days under CK. The juice gathered from passion fruit packaged in PE reached the same amount during 18 days. Ethanol reached up to 0.42 g/100 g within 9 days under no packaging (CK). The fruit obtained from passion fruit packaged in PE reached the same amount in 18 days, with the lowest value observed for the PLA/PBAT sample (0.37 ± 0.09 (mg/100 g) at 18 d). Figure 2c provides the evidence that PLA/PBAT packaging delayed the rising of ethanol content in fruit for over 3 days when compared with the value obtained from PE packaging (12 day for PE and 15 day for PLA/PBAT).

### 2.8. Sensory Evaluation

Figure 2d presents the results of sensory evaluation analysis during 21 d storage, revealing that the score of all groups increased first and then decreased, then the sensory score peaks of CK and packaging groups were reached at 3 and 6 days, respectively. With PLA/PBAT and PE packaging, the flavor fell below the optimum level (score > 20) after 12 days and 9 days, respectively. Thus, packaging materials significantly (*p* < 0.05) reduced the rate of decline in sensory quality during the early and medium storage periods (from day 6 to day 12). The sudden increase in the flavor intensity over time in the early storage period may be due to the increase in the volatile compounds, particularly esters in fruits [24], while the epidermis of the fruit has not yet shrunk due to a large amount of water loss, maintaining a good color. This result demonstrated that, compared with PE, PLA/PBAT can reduce oxygen-free breathing of packaged fruit due to high gas permeability, and hence slow down the odor caused by ethanol production, in agreement with our previous report [19]. The antimicrobial effect of PLA/PBAT packaging film effectively delayed the process of fruit decay and deterioration caused by bacterial and fungal diseases, and the edible quality of the fruit was significantly better than that of other treatments in whole storage.

### 2.9. Intelligent Sensory Evaluation

Figure 3a demonstrates that the variance contribution rates of PC1 and PC2 were 94.63% and 4.29% respectively, and the cumulative contribution rate was 98.92%, which contains most of the electronic nose feature information. The changes of electronic nose responses in unpackaged (CK) and PE-packaged groups could be divided into two stages: 0–12 d and 12–21 d, while the PC1 of electronic nose responses of CK and PE groups changed positively in the first stage, and negatively in the second stage. However, the PC1 of electronic nose responses of the PLA/PBAT group created a slight variation during the whole storage period, which was always in the first stage (0–12 d) sectioning of reference CK and PE. Thus, we can conclude that PAL/PBAT treatment had a smaller change in aroma profile than CK and PE treatment during 21 days, which had no significant difference between the aroma profile at the end of storage and at the medium period of storage (*p* > 0.05).

Figure 3b shows that the variance contribution rates of PC1 and PC2 were 84.01% and 11.75%, respectively, and the cumulative contribution rate was 95.76%, which contains most of the electronic tongue feature information. This was in line with the change of sensory evaluation and electronic nose response signal, indicating that the fruit flavor quality of CK and PE groups became worse suddenly at the 12^th^ day. The PC1 variation distance of CK and PE groups in the first stage (0–12 d) was 0.26 and 0.52 units, respectively, and that of the second stage (12–24 d) was 2.30 and 3.08 units, which is 9.03 and 5.92 times of that in the first stage. PC1 of PAL/PBAT showed a displacement in a positive direction during the whole storage period, which proved that taste profile of passion fruit in PAL/PBAT packaging film had smaller changes than that of CK and PE groups during the storage period of 21 days.

Figure 3c shows the results of principal component analysis of fused information features after homogenizing the response values of probes of e-nose and e-tongue. Variance contribution rates of PC1 and PC2 fused with information characteristics were 87.19% and 9.79%, respectively, and the cumulative contribution rate was 96.98% and over 85%, while PC1 was the main factor contributing to the change of intelligent sensory characteristics processed. In the first stage, PC1 of CK and PE were oriented positive shift from 0 to 12 days, and in the second stage, it changed negatively from 12 to 21 days. The PC1 of PAL/PBAT showed a displacement in a positive direction during the storage period, compared with CK and PE. In conclusion, PLA/PBAT film is more conducive to preserving the overall flavor of passion fruit during storage compared with CK and PE, in agreement with sensory evaluation.

### 2.10. Profile of Volatiles of the Passion Fruit

Table A1 shows that volatile and odoriferous compounds of the passion fruit in different packaging, corresponding to the 0^th^ day, 12^th^ day, and 21^st^ day by GC-MS, contain esters, alcohols, ketones, alkenes, aldehydes, and six other kinds of compounds. A total of 140 flavor components were identified this time. All treatment groups contained ethyl acetate, ethyl propionate, methyl butyrate, ethyl hexanoate, n-hexanol, and methyl heptanone during the whole storage. The dominant volatile substances of passion fruits during the storage period were esters, and the most important representative compounds were ethyl butanoate and ethyl acetate. Packaging materials have great influence on flavor substances of passion fruit during a room-temperature storage period. Ethyl acetate is a colorless clarifying liquid with aromatic odor [25,26]. Ethyl butanone has a clear and intense sweet fruit aroma, with pineapple, banana, and apple flavor, which spread easily but was not lasting, and with fat odor at high concentration [27]. The ethyl acetate concentration of PLA/PBAT-packaged and no-packaged (CK) markedly rose in the 12^th^ day (*p* < 0.05) and reached up to 31.84% and 10.49%. Meanwhile, the content of PE decreased from 24.54% in fresh fruit to 13.76%. The ethyl acetate concentration of PLA/PBAT-packaged and no-packaged (CK) markedly rose in the 12^th^ day (*p* < 0.05), reaching up to 31.84% and 10.49%. Meanwhile, the content of PE decreased from 24.54% in fresh fruit to 13.76%. Fresh passion fruit does not contain ethyl butanoate, which was gradually produced with fruit ripening and senescence. The ethyl butanoate concentration of PE- and PLA/PBAT-packaged markedly rose in the 12^th^ day (*p* < 0.05), reaching up to 20.09% and 43.96%,above CK (29.16%), but there was no significant difference at the end of storage time (21^st^ day) (*p* > 0.05).

Table A2a,b show that there were 39 flavor substances identified in the fresh passion fruit, containing six kinds of esters (14.41%), 17 kinds of terpenoids (64.39%), four ketones (3.55%), two kinds of aldehydes (0.38%), and 10 other classes (17.28%), especially for alcohols, who were detected at the 12^th^ or 21^st^ day. At the 12^th^ day, the ester content of each group increased significantly compared with 0 days, with the highest value observed for the PLA/PBAT sample, while the alcohol substance was significantly lower. At the end of storage time, the ester content of the PAL/PBAT group slightly increased, when compared with 12 days, while the alcohol substance dropped further. In conclusion, PLA/PBAT film is more conducive to preserving the volatile aroma components of passion fruit during storage compared with CK and PE groups, in agreement with sensory evaluation and electronic nose response signals.

## 3. Materials and Methods

### 3.1. Materials

Fresh, mature, purple passion fruit (*Passiflora edulis Sims,* aged 80 days after flowering) was harvested from Xinhongdao Lemon Agriculture Co., Ltd. located in Anyue, Sichuan province, China. The plantation site is at an altitude of 450 m above sea level, with an average temperature of 16.6~18.3 ℃. The fruit was selected for symmetrical spherical shape, weight ranging from 56 to 85 g, and purple color stage (color break) of 70–80 percent, transported to the laboratory in a cooled state, and used for packaging experiments on the same day. Defective fruit (i.e., those showing signs of decay and mechanical damage) were eliminated. Passion fruit of uniform size, color, and shape were consequently used in the above experiments.

In the antibacterial experiment, the packaged films with the same thickness (0.025 mm ± 0.003 mm) were selected with good sales and good reputation in Chinese market at present. PLA/PBAT antibacterial packaging films (food grade) were obtained from Kangrunjie Environmental Protection Technology Co., Ltd. (Jilin, China). PE packaging films (food grade) were obtained from Suzhou Jieda Chemical Plastics Co., Ltd. (Zhejiang, China).

### 3.2. Fruit Preparation and Packaging

Passion fruits were randomly divided into third groups of 70 unit. Two groups were wrapped fruits in PLA/ PBAT- and PE-packaged films, and the third group contained control (unpackaged) passion fruit. Passion fruit in all packaging conditions were sampled immediately after packaging, after 12 h, and then in 3 days intervals thereafter. Packaged and control samples were evaluated after 0, 3, 6, 9, 12, 15, 18, 21 d storage time. All samples were placed into plastic boxes and stored at 20 ± 1 °C and relative humidity (RH) of 70–75 %, a condition based on the common temperature of a display case with air-cooler in most Chinese supermarket. Three packaging samples from each packaging condition were tested, totaling 15 fruit per packaging type. Five fruits were used for chemical quality analysis, and 10 fruits for sensory evaluation.

### 3.3. Quality Evaluation of Storage

The quality of passion fruits was evaluated every two days, for a total of 21 d. Each data was collected from at least three separate experiments. Error bars were used to indicate the estimated error in a measurement or the uncertainty in a value.

#### 3.3.1. Weight Loss Measurements

Weight loss during storage was measured daily using an FA 2004W balance (±0.0001 g; Shanghai Jinghai Instrument Co., Ltd., Shanghai, China) and expressed as (W0 – Wt)/W0, where W0 and Wt are initial weight and tested-date weight, respectively.

#### 3.3.2. Shrinkage Index Measurements

The shrinkage index of passion fruit was characterized every day according to a four-point empirical scale (Table 1) [28].

#### 3.3.3. Total Acid and Total Sugar Measurements

Total acid was determined for 25 g passion fruit juice samples by titration with 0.1 mol/L NaOH solution and expressed as % citric acid [6].

The determination of total sugar was calculated for 20 g passion fruit juice by the method of phenol-vitriol, in which dextrose is the standard sample. The concentration was given in spectroscopy at 490 nm with ultraviolet spectrophotometer (UV-Blue Star A, Beijing Leibotaike Instruments Co., Ltd., Beijing, China). The regression equations of the standard curves based on the concentrations (C) versus the absorptive value (A) are: *y* = 0.0069*x* + 0.0002 (R^2^ = 0.9992).

#### 3.3.4. Firmness Measurements

Firmness was measured at room temperature using a food texture analyzer (FTA) (type TMS-PRO, American TFC Company.), equipped with a 1 kg load cell. A single whole passion fruit was placed on a flat platform and penetrated with a 2 mm diameter flat-head stainless-steel cylindrical probe to a depth of 15 mm at a speed of 1 mm s^−1^, when deformation and minimum induction force were set to 50% and 0.375 N, respectively.

#### 3.3.5. Ascorbic Acid Measurements

Ascorbic acid was determined for 25 g passion fruit juice samples by high-performance liquid chromatography (U3000, Thermo Fisher Scientific, Waltham, MA, USA), in which L ascorbic acid was the universal sample. Ascorbic acid in the sample was drawn with 20 g/L metaphosphate solution by ultrasound for 5 min; a mixture of 1% acetonitrile and 99% of 0.01 mol/L potassium dehydration phosphate (pH 2.5) as the mobile phase. Reduction of L(+)-dehydroretinol acid in the sample by L-cysteine solution were separated on a C18 reverse-phase chromatography column and measured by ultraviolet detector (wavelength = 246 nm). The regression equations of the standard curves based on the concentrations (C) versus the absorptive value (A) are: *y* = 0.1361*x* + 0.0325 (R^2^ = 0.9993)

#### 3.3.6. Ethanol Measurements

Ethanol was determined for 50 g passion fruit juice samples, based on the number of ISO 2148-1973, which is titled “Fruit and vegetable products—Determination of ethanol”.

#### 3.3.7. Sensory Evaluation

The evaluation was performed by 10 trained panels, consisting of five females and five males (age 20–34). All panelists had a professional background in Food Science and Engineering and were trained in evaluation passion fruit, rated from three aspects, as the overall flavor, the degree of pericarp in good condition, and pericarp color [29]. The quality description grade included three grades: extremely poor, medium, and very good, with scores of 1, 5, and 9, respectively, and the specific standards are in Table 2. The summation of three scores was the sensory evaluation total score of the fruit.

#### 3.3.8. E-Nose and E-Tongue Measurements

The juices were marked for 2 g passion fruit juice samples by E-nose (FOX 4000, Alpha MOS, France). This apparatus mainly consists of a sensor array of 18 Metal Oxide Semiconductor (MOS) sensors, a signal collection system, and a data processing software. The details of the sensor array and the process of sampling detection by E-nose have been described by previous reports [30]. For E-noise detection, three samples were prepared for each group.

To detect the liquid fingerprint of 10 g passion fruit juice samples, which were filtered with the filter paper and then diluted into 80 mL with deionized water, an E-tongue (α-Astree, α MOS Company, France) was applied. The E-tongue is mainly made up of an automatic sampler, a sensor array, and a signal processing software. The details of the E-tongue sensor array and the process of sampling detection by E-tongue were previous reported [31]. For E-tongue detection, three samples were prepared for each group.

#### 3.3.9. Aroma of Organic Passion Fruit Measurements

Aroma of organic passion fruit was determined for 2 g passion fruit juice samples by A SQ680 gas chromatograph–mass spectrometer from Perkin Elmer (USA), with an electron impact ionization source (70 eV), used in scan mode. The mass range was from 35 to 400 m/z. The Elite-5MS column (30 m length, 0.25 mm i.d., 0.25 mm film thickness) was maintained at 40 ℃ for 2 min, then programmed to rise to 60 ℃ at 2 °C/min, and held for 1 min. Then programmed to rise to 250 °C at 20 °C/min, and held for 2 min. Helium (99.9999% pure) was used at a flow rate of 1 mL/min, which has a split ratio of 5:1.

The volatile compounds of the passion fruit in different packaging films were identified by GC-MS with Elite-5MS columns, comparing the mass spectra obtained with those of pure standards with those found in the National Institute of Standards and Technology (NIST) library (vers. 1.7), the determination was carried out by combining with the artificial analytical mass spectrometry. The peak area normalization method was utilized to determine the relative content of volatile components.

### 3.4. Statistical Analysis

Multiple samples were tested, and the results were reported as mean ± standard deviation. The obtained values were subjected to analysis of variance, and the means were separated by Duncan’s multiple range test (Super ANOVA, Abacus Concepts, Inc., Berkeley, CA, USA), with *p* values of <0.05 considered significant. The results were treated with principle component analysis (PCA) by SPSS19.0 software.

## 4. Conclusions

In the Chinese market, passion fruit is mainly sold as fresh fruit after ripening, but its pericarp is easily dehydrated and shrinks, which leads to the decline of appearance value and flavor quality. Suitable postharvest packaging would help to spontaneously form a small atmospheric environment, and have a significant impact on shelf-life extension of Passion Fruit by prevented losses from spoilage spreading, water loss, and shrinkage. In Chinese market, the packaging materials widely used in fruits and vegetables are mainly PE-based packaging materials at present. Compared with the blank control, PAL/PBAT and PE films can effectively prolong the shelf life of passion fruit. During passion fruit preservation experiments, PLA/PBAT and PE packaging slowed down the process of fruit metabolism and delayed ripening, as indicated by the concomitant changes of weight loss, shrinkage index, firmness, total sugar, total acid, ascorbic acid content, ethanol content, sensory evaluation, and aroma components. Compared with PE films, PLA/PBAT films are more conducive to preserve the overall flavor of passion fruit during storage time, in agreement with sensory evaluation tested by E-nose, E-tongue, and GC-MS, which also proved that it can effectively maintain the edible quality of passion fruit during storage time. However, compared with PE, PAL/PBAT films did not perform well in maintaining fruit color, shrinkage, and water loss. So the relevant difficulties in technology are still expected to be studied and explored, such as the optimum thickness and storage temperature of PLA/PBAT films.

## Figures and Tables

**Figure 1 molecules-24-03378-f001:**
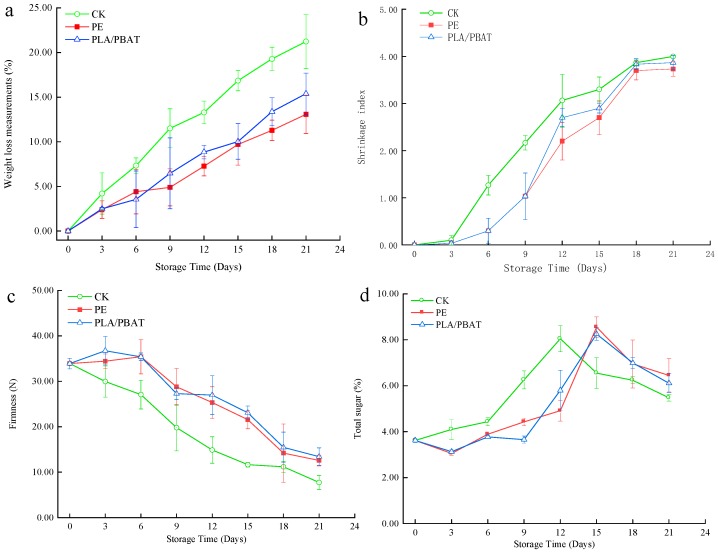
Effect of PLA/PBAT antibacterial packaging film on the (**a**) weight loss, (**b**) shrinkage index, (**c**) firmness, and (**d**) total sugar of passion fruit during storage times (t = 8). PLA: poly(lactic acid); PBAT: poly(butylene adipate-co-terephthalate); CK: unpackaged; PE: polyethylene.

**Figure 2 molecules-24-03378-f002:**
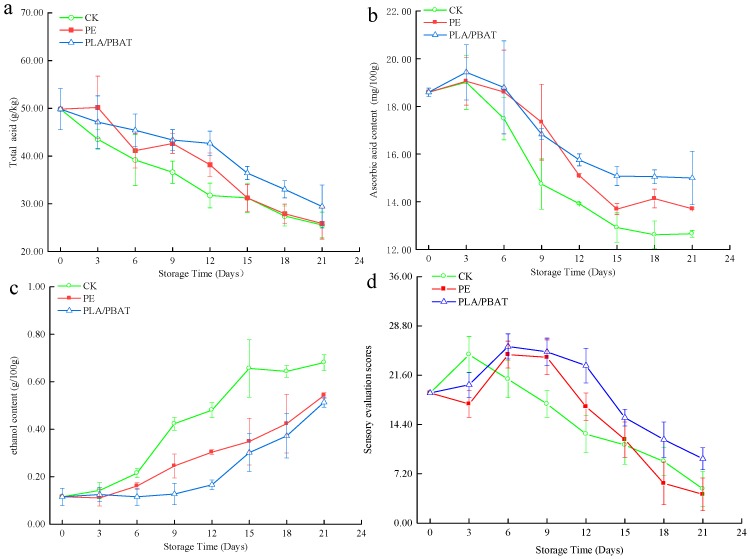
Effect of PLA/PBAT antibacterial packaging film on the (**a**) total acid, (**b**) ascorbic acid, (**c**) ethanol content, and (**d**) sensory evaluation of passion fruit during storage times (t = 8).

**Figure 3 molecules-24-03378-f003:**
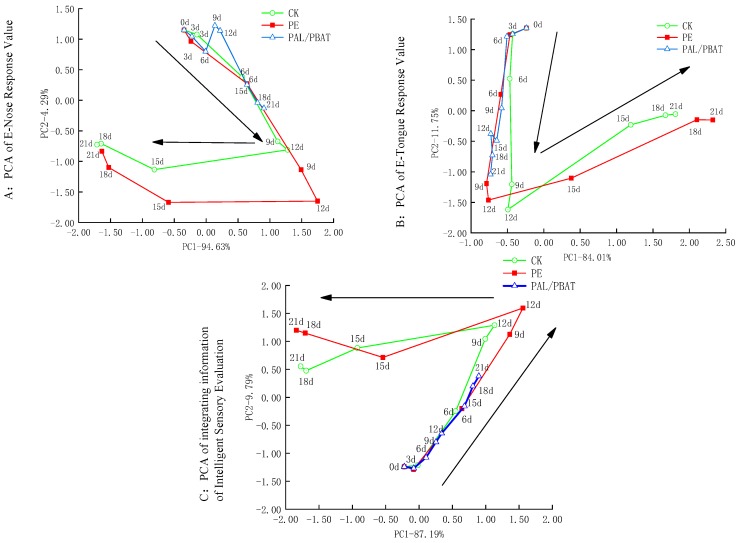
Effect of PLA/PBAT antibacterial packaging film on the (**a**) e-nose response signal, (**b**) e-tongue response signal, (**c**) fusion information of e-nose and e-tongue response of passion fruit during storage times (t = 8), based on principal component analysis (PCA). Note: All black arrows indicate the direction of displacement in PC1.

**Table 1 molecules-24-03378-t001:** Four-point systems used to evaluate the degree of passion fruit shrinkage.

Degree (severity) of Shrinkage	Area of Infected Fruit Surface (%)
0	0
1	0–25
2	25–50
3	50–75
4	>75

The shrinkage index = $\frac{\left(d\times f\right)}{n}$, where d is the category of shrinkage intensity scored on fruit, f is shrinkage occurrence frequency, N is the total number of examined fruit (both healthy and shrinkage).

**Table 2 molecules-24-03378-t002:** The score criterion of sensory evaluation of passion fruit.

Evaluation Item	The Description of Passion Fruit Quality	Score
Overall flavor	Excellent: it has nice taste and sweet smell.	1
	Moderate: it frees from foreign smell, with lighter flavors.	5
	Bad: it tastes too vinegary, or has serious peculiar smell.	9
Degree of pericarp in good condition	Excellent: it has an excellent surface appearance.	1
	Moderate: the pericarp area is more than 30% wrinkled, pitted or rotten.	5
	Bad: the pericarp area is more than 50% wrinkled, pitted or rotten.	9
Pericarp color	Excellent: it has good color and high gloss.	1
	Moderate: it has some slight color variation.	5
	Bad: it has an uneven color distribution and poor gloss.	9

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
