# Peer review of "Effect of PLA/PBAT Antibacterial Film on Storage Quality of Passion Fruit during the Shelf-Life"

_molecules, 2019, doi:10.3390/molecules24183378_

Round 1

Reviewer 1 Report

Dear Authors,

Congratulations to this good paper. I had only one recommendation to the table 3. It is a very long table. I think you can divide this table in two parts:

1) the components (42)  in fresh fruits in the results

2) the components during storage, This table can be added as a supplement, because these components are not discussed

br

The reviewer

Author Response

Comments and Suggestions for Authors 1:

Congratulations to this good paper. I had only one recommendation to the table 3. It is a very long table. I think you can divide this table in two parts:

1) the components (42) in fresh fruits in the results

Re: Revised. Thanks.

2) the components during storage, This table can be added as a supplement, because these components are not discussed.

Re: Yes, you are right, but the academic editor of the Journal suggested us to keep it and added the retention times. At last, I must say sorry to you. But your suggestions make the article better. Thanks a lot.

Reviewer 2 Report

In this manuscript, the storage quality of passion fruit packaged with PLA/PBAT and PE films were compared. Ethanol content, and delay the total acid, ascorbic acid and sensory evaluation were well studied, which is valuable for developing active food packaging material to ensure safety and extend the shelf-life of packaged passion fruit. The volatile and odoriferous compounds of passion fruit in different packaged were investigated by E-nose, E-tongue and GC-MS which fit with the journal. The paper is well-structured, and the reader has no problem to follow the work step by step. However, some issues should be clearly addressed before it can be accepted. The details are listed as the following:
1. If you compare the PLA/PBAT antibacterial packaging film and PE packaging film, you should make sure both the oxygen permeance and water vapor permeability are the same. But in your paper the value seems different, please explain it.
2. In manuscript, you mentioned to test the passion fruit preservation at room temperature. The room temperature usually means 25 °C or 27 °C. Actually, after I read it all I found you only stored it at 20±1°C.
3. There are several types of passion fruit, which type do you used? Purple passion fruit (Passiflora edulis Sims), or yellow passion fruit (Passiflora edulis var. flavicarpa Degenerer) ?
4. How and what kind of package methods do you used ? Vacuum, seal or just wrap it ? Please add the detail.
5. Why you chose 21 d as your storage time ? Some papers reported or your previous studied ? 
6. Line 100: Passion fruits were randomly divided into third groups of 70. Where is the unit of 70 ?
7. I don't think GB T 12456-2008 can be used in the paper. And all the technical information should be in accordance with international methodological standards and guidelines. 
8. Use italic for variables in all the equations. Please make correction to be accurate and consistent through the equation, table and text.
9. Fig. 1 and Fig. 2, it lack of look, layout, and font consistency.
10. In Fig. 3, I don't understand the arrows stand for ? The trend or other else ? Indicate it in legend. 
11. The results reported in this study should be compared with the results reported by López-Vargas Jairo H et al. The following references should be cited. Food Research International Volume 51, Issue 2, May 2013, Pages 756-763. / Meat Science Volume 97, Issue 2, June 2014, Pages 270-276
12. In this paper, the author has done a lot of research on the storage quality of passion fruit, but the conclusion is too simple, this section must be written. 
13. Minor corrections to text, but mostly places where spaces between words are missing, etc.
14. The references must be formatted according to the Journal. 

Author Response

Comments and Suggestions for Authors 2:

In this manuscript, the storage quality of passion fruit packaged with PLA/PBAT and PE films were compared. Ethanol content, and delay the total acid, ascorbic acid and sensory evaluation were well studied, which is valuable for developing active food packaging material to ensure safety and extend the shelf-life of packaged passion fruit. The volatile and odoriferous compounds of passion fruit in different packaged were investigated by E-nose, E-tongue and GC-MS which fit with the journal. The paper is well-structured, and the reader has no problem to follow the work step by step. However, some issues should be clearly addressed before it can be accepted. The details are listed as the following:

If you compare the PLA/PBAT antibacterial packaging film and PE packaging film, you should make sure both the oxygen permeance and water vapor permeability are the same. But in your paper the value seems different, please explain it.

Re: In our experiment, we elaborate selected the two films with good sales and reputation in Chinese market, and make sure the two packaged films with the same thickness (0.025mm±0.003mm). Thanks.

In manuscript, you mentioned to test the passion fruit preservation at room temperature. The room temperature usually means 25 °C or 27 °C. Actually, after I read it all I found you only stored it at 20±1°C.

Re: The reason why we chose the 20±1℃ based on the common temperature of cooler display with air-cooler in supermarket. Thanks.

There are several types of passion fruit, which type do you used? Purple passion fruit (Passiflora edulis Sims), or yellow passion fruit (Passiflora edulis var. flavicarpa Degenerer) ?

Re: Purple passion fruit (Passiflora edulis Sims), and we added the detail in the section of materials. Thanks.

How and what kind of package methods do you used ? Vacuum, seal or just wrap it ? Please add the detail.

Re: Both of the two groups were wrapped with PLA/PBAT or PE packaged films, and the third group is unpackaged passion fruit. Thanks.

Why you chose 21 d as your storage time ? Some papers reported or your previous studied ?

Re: The reason why we chose 21 day is based on our previous experiment. Under this temperature, the passion fruit lost its commercial value after 3 weeks of storage. Thanks.

Line 100: Passion fruits were randomly divided into third groups of 70. Where is the unit of 70 ?

Re: Added. Thanks.

I don't think GB T 12456-2008 can be used in the paper. And all the technical information should be in accordance with international methodological standards and guidelines.

Re: Changed. The key technologies are discussed. Ascorbic acid was determined for 25 g passion fruit juice samples by high performance liquid chromatography (U3000, Thermo Fisher Scientific, American), in which L ascorbic acid was the universal sample. Ascorbic acid in the sample was drawn with 20g/L metaphosphate solution by ultrasound for 5 minutes. A mixture of 1% acetonitrile and 99% 0.01mol/L potassium dehydration phosphate (pH 2.5) as the mobile phase. Reduction of L(+)-dehydroretinol acid in the sample by L-cysteine solution were separated on a C18 reverse-phase chromatography column, and measured by ultraviolet detector (wavelength = 246nm). The regression equations of the standard curves based on the concentrations (C) versus the absorptive value (A) are: y = 0.1361x - 0.0325 (R2 = 0.9993). Thanks.

Use italic for variables in all the equations. Please make correction to be accurate and consistent through the equation, table and text.

Re: Revised. Thanks.

Fig. 1 and Fig. 2, it lack of look, layout, and font consistency.

Re: Done. Thanks.

In Fig. 3, I don't understand the arrows stand for ? The trend or other else ? Indicate it in legend.

Re: All arrows stand for the direction of displacement in PC1.

The results reported in this study should be compared with the results reported by López-Vargas Jairo H et al. The following references should be cited. Food Research International Volume 51, Issue 2, May 2013, Pages 756-763. / Meat Science Volume 97, Issue 2, June 2014, Pages 270-276

Re: The references were added. Thanks.

In this paper, the author has done a lot of research on the storage quality of passion fruit, but the conclusion is too simple, this section must be written.

Re: The conclusion was re-written. Thanks.

Minor corrections to text, but mostly places where spaces between words are missing, etc.

Re: Done. In the Chinese market, Passion fruit is mainly sold as fresh fruit after ripening, but its pericarp is easily dehydrated and shrinks, which leads to the decline of appearance value and flavor quality. Suitable postharvest packaging would help to form small atmospheric environment spontaneous, and have a significant impact on shelf-life extension of Passion Fruit by prevented losses from spoilage spreading, water-loss and shrinkage. In the Chinese market the packaging materials widely used in fruits and vegetables are mainly PE-based packaging materials at present. Compared with the blank control, PAL/PBAT and PE packaging can effectively prolong the shelf life of passion fruit. During the room temperature passion fruit preservation experiment, PLA/PBAT and PE packaging slowed down the process of fruit metabolism and delayed ripening, as indicated by the concomitant changes of weight loss, shrinkage index, firmness, total sugar, total acid, ascorbic acid content, ethanol content, sensory evaluation and aroma components. Compared to PE films, PLA/PBAT films is more conducive to preserve the overall flavor of passion fruit during storage time, in agreement with sensory evaluation tested by E-nose, E-tongue and GC-MS, which also proved that it can effectively maintain the edible quality of passion fruit during storage time. However, compared with PE, PAL/PBAT film did not perform well in maintaining fruit color, shrinkage and water loss. The result shows that the relevant difficulties in technology are still expected to be studied and explored, such as the optimum thickness and storage temperature of PLA/PBAT films. Thanks.

The references must be formatted according to the Journal.

Re: Done. Thanks.

Reviewer 3 Report

Paper describes interesting data of PLA-PBAT antibacterial film on storage 
quality of passion fruit during the shelf-life. I found this topic very well, but quality of presented data is low. Volatiles presented on Table 3 are not supported with data.

Please present retention indices (in Table 1 are wrongly presented); Please show calculated RI as well as their comparision with NIST or other literature data; Cyclopropyl isothiocyanate is missinterpreted; Please re-check RI with lirt. as well as MS spectra; (3R)-(+)-Isosylvestren is missinterpreted; Please re-check RI with lirt. as well as MS spectra; Linalyl anthranilate is missinterpreted; Please re-check RI with lirt. as well as MS spectra; Instead of (4-methyl-1-propan-2-ylcyclohex-3-en-1-yl) acetate use Terpinene 4-acetate; 3-methyldodec-1-yn-3-ol is missinterpreted;  Please re-check RI with lirt. as well as MS spectra; 4-Methylcyclohexanol  is missinterpreted;   Please re-check RI with lirt. as well as MS spectra; [(2R,5S)-5-Methyl-5-vinyltetrahydro-2-furanyl]-2
propanol is missinterpreted;   Please re-check RI with lirt. as well as MS spectra; Trans-4-Methylcyclohexanol Please re-check RI with lirt. as well as MS spectra; 1,3,5,5-TetraMethyl-1,3-cyclohexadiene  is missinterpreted;   Please re-check RI with lirt. as well as MS spectra; 3,4-dimethylocta-2,4,6-triene is missinterpreted;   Please re-check RI with lirt. as well as MS spectra; 1-methylidene-4-prop-1-en-2-ylcyclohexane is missinterpreted; Please re-check RI with lirt. as well as MS spectra; 1-Cyclopropylethanone is missinterpreted;   Please re-check RI with lirt. as well as MS spectra; Cyclopentanone is missinterpreted;   Please re-check RI with lirt. as well as MS spectra; Cyclopentenone is missinterpreted;   Please re-check RI with lirt. as well as MS spectra; 2-butan-2-ylcyclopentan-1-one is missinterpreted;   Please re-check RI with lirt. as well as MS spectra; 3-prop-2-enylidenecyclobutene is missinterpreted;   Please re-check RI with lirt. as well as MS spectra; What is Ch3cho? Ammonium carbonate is artefact? Hexamethylcyclotrisiloxane is from column; Decamethylcyclopentasiloxane is from column; Propylene oxide could be missinterpretation; Nα,Nω-Dicarbobenzoxy-L-arginine is missinterpreted;   Please re-check RI with lirt. as well as MS spectra; 4-ethenyl-1,2-dimethylbenzene could missinterpreted;   Please re-check RI with lirt. as well as MS spectra; 4-Methyl-2-(2-methylprop-1-en-1-yl)tetrahydro-2H-pyran  use common name for rose oxide; 2-Ethenyl-1,1-dimethyl-3-methylenecyclohexane could missinterpreted;   Please re-check RI with lirt. as well as MS spectra; Azoxymethane could be missinterpreted;   Please re-check RI with lirt. as well as MS spectra; remove stereochemical marks from all interpreted compounds

Author Response

Comments and Suggestions for Authors 3:

Paper describes interesting data of PLA-PBAT antibacterial film on storage quality of passion fruit during the shelf-life. I found this topic very well, but quality of presented data is low. Volatiles presented on Table 3 are not supported with data.Please present retention indices (in Table 1 are wrongly presented); Please show calculated RI as well as their comparision with NIST or other literature data;

Re: The volatile compounds of the passion fruit in different packaging films were identified by GC-MS with Elite-5MS columns, comparing the mass spectra obtained with those found in the NIST library (vers. 1.7), the determination was carried out by combining with the artificial analytical mass spectrometry. The peak area normalization method was utilized to determine the relative content of volatile components. Because this test of GC-MS belonged to sample test, and the testing platform did not take pure standard reference material such as n-hexane, so there is no retention index (RI) obtained, but only the retention time (RT)       in the original data provided as a qualitative reference. Your opinion is great, I will further improve the detection method in the following experiments. Thanks.

Instead of (4-methyl-1-propan-2-ylcyclohex-3-en-1-yl) acetate use Terpinene 4-acetate;

Re: Revised. Thanks.

What is Ch3cho?

Re: Corrected, it is acetaldehyde, Thanks.

Ammonium carbonate is artefact?

Re: There was no ammonium carbonate. Thanks.

Hexamethylcyclotrisiloxane is from column; Decamethylcyclopentasiloxane is from column;

Re: Deleted. Thanks.

4-Methyl-2-(2-methylprop-1-en-1-yl)tetrahydro-2H-pyran use common name for rose oxide;

Re: Revised. Thanks.

[(2R,5S)-5-Methyl-5-vinyltetrahydro-2-furanyl]-2 propanol is missinterpreted; Please re-check RI with lirt. as well as MS spectra;

Re: Yes, I checked this component is [(2R,5S)-5-Methyl-5-vinyltetrahydro-2-furanyl]-2 propanol, which is one of the essential oils in plant used for flavor formulations.

Cyclopropyl isothiocyanate is missinterpreted; Please re-check RI with lirt. as well as MS spectra; (3R)-(+)-Isosylvestren is missinterpreted; Please re-check RI with lirt. as well as MS spectra; 4-Methylcyclohexanol is missinterpreted; Please re-check RI with lirt. as well as MS spectra; 1,3,5,5-TetraMethyl-1,3-cyclohexadiene  is missinterpreted;   Please re-check RI with lirt. as well as MS spectra; 3,4-dimethylocta-2,4,6-triene is missinterpreted;   Please re-check RI with lirt. as well as MS spectra; 1-Cyclopropylethanone is missinterpreted;   Please re-check RI with lirt. as well as MS spectra; Cyclopentanone is missinterpreted;   Please re-check RI with lirt. as well as MS spectra; Cyclopentenone is missinterpreted; Please re-check RI with lirt. as well as MS spectra; 2-butan-2-ylcyclopentan-1-one is missinterpreted;   Please re-check RI with lirt. as well as MS spectra;Nα,Nω-Dicarbobenzoxy-L-arginine is missinterpreted;   Please re-check RI with lirt. as well as MS spectra;4-ethenyl-1,2-dimethylbenzene could missinterpreted;   Please re-check RI with lirt. as well as MS spectra; Azoxymethane could be missinterpreted;   Please re-check RI with lirt. as well as MS spectra; remove stereochemical marks from all interpreted compounds ;Linalyl anthranilate is missinterpreted; Please re-check RI with lirt. as well as MS spectra; 3-methyldodec-1-yn-3-ol is missinterpreted;  Please re-check RI with lirt. as well as MS spectra; Trans-4-Methylcyclohexanol Please re-check RI with lirt. as well as MS spectra; 3-prop-2-enylidenecyclobutene is missinterpreted;   Please re-check RI with lirt. as well as MS spectra; Propylene oxide could be missinterpretation; 2-Ethenyl-1,1-dimethyl-3-methylenecyclohexane could missinterpreted;   Please re-check RI with lirt. as well as MS spectra;

Re: We deleted the following components after double-checked the original data files and further literatures: Cyclopropyl isothiocyanate, (3R)-(+)-Isosylvestren, 4-Methylcyclohexanol, Trans-4-Methylcyclohexanol, 3,4-dimethylocta-2,4,6-triene, Cyclopentenone, 1-Cyclopropylethanone, 2-butan-2-ylcyclopentan-1-one, Nα, Nω-Dicarbobenzoxy-L-arginine, 4-ethenyl-1, 2-dimethylbenzene, Azoxymethane, Linalylanthranilate, 3-methyldodec-1-yn-3-ol, Trans-4-Methylcyclohexanol, 3-prop-2-enylidenecyclobutene, Propylene oxide, 2-Ethenyl-1, 1-dimethyl-3-methylenecyclohexane and Epoxypropane. A total of 140 flavor components were identified this time. Based on the above results, Table 3 was replaced. Thanks.

Round 2

Reviewer 3 Report

Thanks for improvement of this version of manuscript. Some of compounds were removed. I still recommend to calculate retention indices (RI) for individual compounds and comparison thair values with presented data in NIST, Adam's or other library. Injection of n-alkanes series  (easy commercially available) could make the interpretation more undoubtedly.

Author Response

Comments and Suggestions for Authors 3:

Thanks for improvement of this version of manuscript. Some of compounds were removed. I still recommend to calculate retention indices (RI) for individual compounds and comparison thair values with presented data in NIST, Adam's or other library. Injection of n-alkanes series (easy commercially available) could make the interpretation more undoubtedly.

Re: Thanks for your suggestions. In our subsequent studies, such as qualitative and quantitative analysis of volatile substances of Passiflora packaged with PLA/PBAT, injection of n-alkanes series could make the interpretation more undoubtedly. But now, we have to explain the reasons why we believe it is no need to calculate retention indices (RI) for individual compounds. Firstly, the aroma feeling is one of the most important quality for fresh-fruit that determines consumption desire. The core purpose of aromas of passion fruit measurements is to prove the obvious difference of odor caused by the three packaging treatments in the sensory analysis represented by electronic noise in this paper. Secondly, this part of analysis is focus on the differences kinds of volatile gas in the three packaging treatments at different storage time. Even if there was no retention index for the injection of pure n-alkanes, it can still be demonstrated by comparing the results of the NIST library. Finally, in order to increase the credibility of the results, we added the match and R. Match for all components in Table A3. Thank you again for your efforts to make the paper better.